# The Psychosocial Model of Absenteeism: Transition from 4.0 to 5.0

**DOI:** 10.3390/bs13040332

**Published:** 2023-04-14

**Authors:** Aleksandra Sitarević, Ana Nešić Tomašević, Aleksandar Sofić, Nikola Banjac, Nenad Novaković

**Affiliations:** Faculty of Technical Sciences, University of Novi Sad, Trg Dositeja Obradovića 6, 21000 Novi Sad, Serbia

**Keywords:** absenteeism, job characteristics, mental health, Industry 4.0, Industry 5.0

## Abstract

The research provides insight into the factors that determine absenteeism in different types of organizations, in order to facilitate the adjustment of employees and organizations in the transition process from Industry 4.0 to Industry 5.0. The aim of the study is to predict the absenteeism of employees in the context of job characteristics and mental health. Additionally, the research investigated the effect of size, ownership, and sector of the companies on absenteeism, job characteristics, and mental health. The sample included responses from 502 employees of different sociodemographic characteristics that work in various types of organizations, performing white-collar and blue-collar jobs. A short mental health questionnaire—Mental Health Inventory, MHI-5—was used in order to measure mental health. The Job Characteristics Questionnaire was used to measure the employees’ perceptions of their job characteristics—job variety, autonomy, feedback, dealing with others, task identity, and friendship. The absenteeism is operationalized with the question: “During the past 12 months, how many days were you absent from work for any reason?”. The findings suggest that mental health and job characteristics significantly reduce absenteeism among different sectors. The result showed that the size, ownership, and sector of the organization significantly affect the absenteeism, job characteristics, and the mental health of the employees. The results support the premises of Industry 5.0 and offer a new human-centric approach to absenteeism through the promotion of mental health through long-term organizational strategies and a more inclusive approach to employees’ preferences in relation to job characteristics. The study offers a new, double-sided model of absenteeism, determining causal factors from the perspective of personal and organizational factors.

## 1. Introduction

Industry 4.0 has shown the world the remarkable progress of digital technologies, ensuring the more rapid process of globalization through new technologies such as Big Data, Artificial Intelligence (AI), Cloud Computing (CC), Internet of Things (IoT), Internet of Services (IoS), robotics, and 3D printing [1]. These technologies transformed the working environment entirely, producing new threats to business and society [2]. The effects of new disruptive technologies became even more conspicuous with the circumstances of the COVID-19 pandemic, which transformed and limited the working environment [3], affecting the employee’s well-being [4]. A new working context unequivocally leads to counterproductive behaviors [5] that are challenging to manage, especially considering that work is one of the most significant determinants of quality of life [6]. Industry 5.0, promotes sustainable working conditions supposing that will boost the potential of workers and assure better performance of the organizations [7,8]. Additionally, the new demands on employees are based on positive expectations that mentally healthy employees have a need for job variety, autonomy, feedback, dealing with others, task identity, and friendship, to the extent to which they feel good. Previous studies have shown that extensive and unfavorable working environments can raise challenges in physical and social functioning among employees of different occupations [9,10]. Chronic conditions of the employees could potentially jeopardize the subsistence of their companies due to loss of productivity and higher rates of absenteeism [11], with the severe economic burden. Table 1 below provides data on the annual costs associated with absenteeism among different occupations [12].

Absenteeism is a strategically relevant problem in different sectors [13,14,15,16] and types of organizations [17,18]. During COVID-19, the absence of employees was one of the most challenging organizational behaviors among different occupations. The crisis has had an extensive effect on employees, affecting their psychosocial functioning and their productivity at work. Despite the fact that the pandemic has inevitably transformed working conditions, absenteeism in companies was challenging for HRM long before the pandemic situation. Generally, absenteeism, especially when unplanned, could trigger disruptions that might have irreversible consequences for job performance, especially in a labor-intensive setting. Managing absenteeism is one of the main challenging actions in HRM that an organization may pursue in order to assure company success [19]. The causes of absenteeism can be determined by demographic, personal, and organizational factors. At the individual level, demographic and personal variables indicate contradictory results [20,21,22,23]. In this sense, different models of absenteeism emphasize the importance of work characteristics [23], especially in large corporate systems. Absenteeism, turnover, and job dissatisfaction are considered the leading problems of poor organizational performance, so one of the focuses of human resource management should be on designing job characteristics in the workplace with the aim of creating a dedicated workforce. There is a lack of evidence that determine absenteeism in relation to the mental health status of the employees and job characteristics among different types of organizations. Additionally, this paper examined the effect of size, ownership, and sector of the organizations on the variables included in the model. According to the literature gaps, the authors proposed the psychosocial model of absenteeism as a new perspective among employees who work in different types of organizations in terms of size, ownership, and occupation. The research should answer the labor challenges by proposing evidence-based predictive solutions in terms of reducing absence by designing the work environment and boosting the mental health of employees through the business strategy. The aim of the study is to predict the absenteeism of employees in the context of job characteristics and mental health. To investigate absenteeism in the organizations, the authors propose the following conceptual framework in Figure 1.

Based on the previous mentioned, the authors proposed the following research questions:RQ1: Do the level of mental health and job characteristics statistically significantly predict absenteeism among the employees?RQ2: Does the size of the organization have an effect on the rate of absenteeism, mental health, and job characteristics?RQ3: Does organizational ownership have an effect on absenteeism rates, mental health, and job characteristics?RQ4: Does the sector of organization in which employees work have an effect on the rate of absenteeism, mental health, and job characteristics?

The paper is structured as follows: Section 1 presents the main findings from the literature. Section 2 describes the quantitative method approach with the sample, measures, and data collection procedure. Section 3 presents the results of the paper. Further, Section 4 describes the practical and theoretical implications of the study. At the end, Section 5 summarizes the conclusions, limitations, and further implications of the research topic.

### 1.1. Absenteeism

Absenteeism is still one of the most significant problems in organizations. It is defined as absence from work at the time scheduled for performance. On the other hand, sick leave as a more specific category of absenteeism refers to absence from work due to health problems [24]. Absenteeism is a complex phenomenon that has multiple individual, social, and financial impacts, so it is an organizational behavior that leads to a wide range of counterproductive behaviors of employees [22]. The literature is devoting attention to absenteeism to determine the factors that might have an effect on it, including a large number of variables. Previous research has shown a clear and unambiguous relation between absenteeism and certain aspects of health [25,26,27,28]. However, it has not been clearly established whether different mental states of employees contribute more to absenteeism or whether absences are solely due to physical illness or disability. Moreover, various job characteristics significantly affect the level of absenteeism [29,30]. Previous research found that large companies have higher levels of absenteeism compared to companies of small size [31]. This relationship between absenteeism and organization size could be explained by the fact that large companies have lower unit costs of absenteeism. In addition to this, it is assumed that larger companies represent a more suitable context for absence from work, due to highly developed procedures and rules, where more workers are specialized in one job, while this is often not the case in smaller companies; therefore, the work directly suffers and often stops. Mental health issues are thought to affect small businesses to a much greater extent than larger businesses. Due to their size and fewer resources, SMEs cannot afford to have employees not working at full capacity. Evidence from many countries shows that public-sector employees have significantly higher absenteeism rates than private-sector employees [32]. On the other hand, findings indicate that employees in the private sector have better mental health than their colleagues in the public sector [33].

### 1.2. Job Characteristics

Job characteristics theory [34] describes the relationship between certain job characteristics and individual responses to the work of employees and it represents the work design approach. Sims and colleagues [35] proposed a model of job characteristics operationalized by friendship, skill variety, autonomy, feedback, work identity, and cooperation. The dimension of friendship is indicating the possibility to build informal relationships with colleagues, and it is perceived as fostering a supportive environment that increases organizational outcomes such as organizational commitment and job satisfaction [36,37]. Skill variety is related to the range of activities and talents that the job is demanding; it is positively correlated with meaningfulness to work, job satisfaction, motivation, involvement at work, and with high compensation expectations [37]. Autonomy is the level of freedom that workers can obtain in terms of methods and scheduling with respect to performing work. Mental issues and conflicts at work are significantly decreased by giving more autonomy to employees, and mental well-being has a positive effect on the perception of work motivation and job satisfaction [38]. The next job characteristic is feedback which measures the level of information that a worker receives about his performance on a specific job [39]. In the terms of this model, friendship is related to the level of one’s ability to build informal relationships at work. Identity indicates and uniqueness level and clarity of a job assignment under the assumption that the overall work is made up of meaningful parts [40]. The last job characteristic of the model is dealing with others which measures the degree to which a certain job requires employees to deal with other people in order to complete the work from beginning to end [35]. Job characteristics represent a significant factor that affects the work motivation of employees by achieving specific psychological states [41]. Yahya [42] states that in addition to the optimal level of primary competencies of employees needed to perform the job, in terms of knowledge, skills, abilities, and behavior, and contextual factors, such as job characteristics, play an important role; they are equally important as the primary competencies of employees in providing high-performance level. Therefore, both factors (i.e., personal and contextual) are dynamic in nature and need to be integrated in a compatible manner, which is often very challenging for human resource management.

### 1.3. Mental Health

The World Health Organization [43] is defining mental health as “a state of well-being in which an individual realizes his or her own abilities, can cope with the normal stresses of life, can work productively and fruitfully, and is able to make a contribution to his or her community”. Mental health includes the absence of psychopathology, in terms of anxiety and depression, as well as the presence of psychological well-being. In the occupational context, mental issues are affecting the functioning of the employees which might lead to economic costs due to lower employee productivity and higher rates of absenteeism [44]. Schultz and colleagues [45] determined the direct effects of health and mental well-being on business productivity, and their results indicated that the costs of absenteeism are related to the level of poor health conditions of workers. Wee and colleagues [46] found that a combination of socioeconomic, physical, and mental health factors predicted absenteeism among different employee demographics. Bailey and colleagues [47] hypothesized that mental health difficulties impair worker functioning and work performance, which is mainly reflected in absenteeism. Walker and Bamford [48] found that frequent employee absences due to chronic health problems could significantly affect business productivity, and they strongly recommended that managers must balance the need to maximize productivity with the needs of employees experiencing health problems [49]. On the other hand, health promotion programs should not be implemented only as a strategy to maximize the organization’s profit margin but should be implemented in the context of social justice that contributes to improving the well-being and quality of life of people in and outside the workplace [46]. The contemporary insights into problems that occur with mental health issues have opened key questions for organizations and society as a whole in terms of approaching to mental health [50]. The ongoing Industry 4.0, with its focus on efficiency and performance, pushed the feelings and mental capacities of employees through digitization and the introduction of new technologies [51]. Despite the fact that new technologies shaped the working environment, especially in terms of reducing the intensity especially in the context of manual work, the new working patterns with higher working requirements brought challenges and limitations that could endanger the mental health of the employees, indicated in terms of discomfort and extreme emotions at work affecting their working performance and absence. Realizing the importance of the challenges that Industry 4.0 faces, the European Commission [52] emphasized that the mental health of workers is a key factor in the implementation strategy of Industry 5.0, aimed at achieving a sustainable, inclusive, and more resilient working system that aligns with Society 5.0 [53].

## 2. Materials and Methods

### 2.1. Sample

The sample included responses from 502 employees of different sociodemographic characteristics that work in different types of organizations, performing white-collar and blue-collar jobs. The sample structure is shown in the Table 2 below.

### 2.2. Measures

A short mental health questionnaire—Mental Health Inventory, MHI-5 [54]—was used in order to measure mental health. The original version of the questionnaire consists of five questions intended to measure general mental health. The parts of the questionnaire cover the basic domains of mental health, and the respondent’s task is to estimate the frequency for each item (from 1 = always to 6 = newer). The instructions emphasize that the respondents should estimate the item for the period of the past month. Domains of the questionnaire include anxiety, depression, general positive affect, and behavioral/emotional control. Response format for all items are 1 = strongly disagree; 3 = not sure, and 5 = strongly agree. The total score is defined as the sum of the results on all items, whereby a higher score indicates a higher level of general mental health.

The Job Characteristics Questionnaire [35] is used as a measure of employees’ perceptions of their job characteristics.The Questionnaire measures six job characteristics: variety, autonomy, feedback, interaction with others, task identity, and friendship. Previous research has the shown high reliability of individual dimensions and the total score. For items 1–13, the response formats are 1 = strongly disagree, 3 = not sure, and 5 = strongly agree. For items 14–30, the item format is 1 = minimum amount, 3 = moderate amount, and 5 = maximum amount.

The questionnaire for assessing absenteeism at the workplace is operationalized with the question: “During the past 12 months, how many days were you absent from work for any reason?”.

Sociodemographic variables:Gender—defined in two categories: male and female;Age—defined as a numerical variable as the number of years;Ownership of the company—defined through the following categories: private or public;Size of the organization—defined through the following categories: small, medium, and large;Sector of the organization—defined through the following categories: health, hospitality, administration, production.

### 2.3. Data Collection

The sample was collected in paper–pen form in cooperation with the HR sectors of the organizations where the research was planned to be conducted. Before the research was conducted, each respondent was asked to participate and sign a formal informed consent form voluntarily. All participants were asked to be sincere in giving their responses. After computing the scores, the scales were presented by descriptive statistics (measures of central tendency and measures of variability). The reliability of the scales was investigated by the internal consistency (Cronbach’s alpha). Multiple linear regressions were used to investigate the effect of predictors on criteria variable. *t*-test was used to investigate the differences in absenteeism, job characteristics, and mental health between the employees that have worked in companies of different ownership. Multivariate analysis of variance (MANOVA) was used to investigate the effects of size, and sector of organizations on absenteeism and job characteristics. For significant effects, Scheffe’s post hoc test was used. The corrected model of F test was used in case when Box’s test showed that equality of covariance were violated. 

## 3. Results

Table 3 below shows the results of descriptive statistics and the reliability of the constructs included into the research—mental health, job characteristics, and absenteeism. It can be determined that the mental health of the employees is moderate. Friendship is the highest expressed job characteristic. The absenteeism average is 24 days. Reliabilities of applied scales and subscales are satisfactory (α > 0.60).

The results below are investigating whether the level of mental health and job characteristics significantly predict absenteeism in a sample of employees in organizations of different sectors. Four regression model were defined, one for each sector—hospitality (R^2^ = 0.109, F(7,89) = 20.56, *p* < 0.05), health (R^2^ = 0.085, F(7,95) = 1.26, *p* > 0.05), production (R^2^ = 0.145, F(7,80) = 20.07, *p* < 0.05), and administration (R^2^ = 0.112, F(7,206) = 4.35, *p* < 0.05). The results of the multiple linear regression determined that mental health is the stable negative predictor of absenteeism among the organizations in all four included sectors. The prediction of absenteeism in the context of job characteristics gives a slightly different picture. Autonomy is a positive predictor of absenteeism in the hospitality sector, feedback is a negative predictor of absenteeism in administration, while friendship is a negative predictor of absenteeism in the administration sector.

The text below shows the results of multivariate tests indicating the significant effect of different company sizes and the expression of mental health, job characteristics, and absenteeism among the employees. There were significant differences between groups of organizations of different sizes (Wilks’ Lambda = 0.92, F(16,984) = 2.52, *p* = 0.001; η_p_^2^ = 0.039). Tests of the between-subjects effect showed that differences exists in Job variety (F(2,2.588) = 7.07, *p* = 0.001, η_p_^2^ = 0.028), Dealing with others (F(2,2.941) = 5.79, *p* = 0.003; η_p_^2^ = 0.023), Friendship (F(2,1.936) = 3.96, *p* = 0.020; η_p_^2^ = 0.016), and Mental health F(2,3.650) = 5.36, *p* = 0.005; η_p_^2^ = 0.021). The results of Scheffe’s post hoc tests showed the significant differences in Job variety, Dealing with others, Friendship and Mental health (for mean scores see Table 4). The rest of the tests were not significant (*p* > 0.05). The results of Scheffe’s post hoc test are presented in the Table 5 and indicate that job variety is more expressed in large companies than in the small (*p* = 0.012) and medium (*p* = 0.012). Dealing with others is more expressed in large companies than in the small (*p* = 0.030) and medium (*p* = 0.023). Friendship is higher in large than in medium size organizations (*p* = 0.040). In the end, mental health is more expressed among the employees that work in small organizations than in large (*p* = 0.005).

The results of the *t*-test on independent samples indicate that there are significant differences in job characteristics in companies of different ownership. Specifically, the level of Autonomy is a characteristic in private-owned companies, while Dealing with others and Friendship are more pronounce in public companies. The ownership of the company is also the significant factor of the employees’ mental health and absenteeism rate. Higher rates of mental health and absenteeism are determined among the employees in public companies.

The results of multivariate tests indicate the significance of differences between the employees that work in different companies in different sectors and the expression of mental health, job characteristics, and absenteeism. There were significant differences between groups of employees that worked in organizations of different sectors (Wilks’ Lambda = 0.76, F(24,1424.51) = 5.46, *p* = 0.000, η_p_^2^ = 0.082). Tests of the between-subjects effect showed that differences exists in Job variety (F(3,1.503) = 4.08, *p* = 0.007, η_p_^2^ = 0.024), Feedback (F(3,5.933) = 8.11, *p* = 0.000, η_p_^2^ = 0.047) Dealing with others (F(3,3.927) = 7.90, *p* = 0.000; η_p_^2^ = 0.045), Friendship (F(3,2.271) = 4.69, *p* = 0.003; η_p_^2^ = 0.027), and Mental health F(3,7.374) = 11.289, *p* = 0.000; η_p_^2^ = 0.064). The results of Scheffe’s post hoc tests showed the significant differences in Job variety, Feedback, Dealing with others, Friendship, and Mental health (for mean scores see Table 6). The rest of the tests were not significant (*p* > 0.05). The results of Scheffe’s post hoc test are presented in the Table 7 and indicate that job variety is more expressed in health sector than in administration (*p* = 0.043). Feedback is more expressed in production (*p* = 0.000) and in administration (*p* = 0.000) than in the health sector. Dealing with others is more expressed in the health sector than in administration (*p* = 0.024), and more expressed in production than in hospitality (*p* = 0.009) and administration (*p* = 0.001). Friendship is more expressed in production than in the administration (*p* = 0.004). Higher levels of mental health are determined in the sector of hospitality than in health (*p* = 0.020) and production (*p* = 0.001). Finally, higher levels of mental health are determined in the administration sector than in health (*p* = 0.002) and production (*p* = 0.000).

## 4. Discussion

The aim of the research is to investigate the factors that affect the rate of absenteeism at work. The proposed model included the level of mental health of employees and job characteristics of the workplace as supposed significant predictors. Further, the effect of different determinants of organizations such as size, ownership, and sector on absenteeism, job characteristics, and mental health was investigated. Additionally, the application of the MHI-5 questionnaire in different sectors is of great importance for understanding the mental health of workers and the need for interventions to improve the mental health and well-being of workers. Previous research from a similar cultural area showed favorable metric characteristics in terms of reliability on the general population of respondents [55]; however, this research is considered a pioneer in terms of obtaining data on the mental health of employees in different sectors.

Absenteeism refers to a worker’s absence from work for different reasons. The Labour Law [56] in the Republic of Serbia recognizes the following reasons for employee absence: annual leave, national holidays, religious holidays, paid leave, sick leave, maternity leave, military service, suspension, unpaid leave, resignation period, work-related injury, and voluntary leave. As in the literature review [29,30,46], the proposed model showed that the level of mental health and job characteristics statistically significantly predict absenteeism among different sectors of the organization.

The new demands of Industry 5.0 on employees in different sectors imply different job designs in terms of job variety, autonomy, feedback, dealing with others, task identity, and friendship to the extent to feel comfortable [57,58] and reduce the possibility of unplanned absenteeism [23,29,30,59].

The results of this research have shown that mental health is a stable, negative predictor among employees in all included sectors, but the effects of different job characteristics in terms of absenteeism are not so consistent. These relations are emphasizing that occupational context must be taken into account in job design [60] by restricting the range of expression of certain job characteristics.

In the sector of hospitality, the authorization level should be lower in order to produce a comfortable and motivating climate with a low rate of absence. In the sector of administration and production, in terms of job design, giving information about the performed work and emphasizing informal social ties are preferable. The results are indicating that feedback culture could keep employees in the workplace. The results confirmed the supposed effect of size on mental health and the expression of different job characteristics, but not in absenteeism, as previous studies have shown that smaller companies have the higher rates [31]. Job performing in larger companies requires more variety and different activities in carrying out the work and includes a larger number of different skills and talents of employees compared to SMEs. Due to smaller business systems, the employees emphasized dealing with others and friendship as more expressed job characteristics compared to larger companies. In a similar way, SMEs are sharing ties of social interaction in performing work. Mental health is more expressed among the employees that work in small organizations than in large ones, as the previous studies have shown [31]. Mental health issues more frequently affect individuals employed in small companies, compared to larger business systems. Due to their size and fewer resources, SMEs cannot afford to have employees not working at full capacity, so HRM should pay special attention to establishing a sustainable and pleasant working environment for its employees. Public and private companies differ in levels of absenteeism, job characteristics, and mental health, as was shown in previous studies [32,34,61]. It seems that private ownership provides conditions for employees in which they do have a choice and more control over the job they are doing compared to public organizations. In the context of the large bureaucratic environment, public companies might unconsciously promote learned helplessness in which employees are not motivated and are losing the initiative for further jobs because, in these circumstances, they are aware that they do not have control over the outcomes of their decisions [62]. On the other hand, social ties, such as dealing with others to successfully complete the job and making informal relationships with colleagues, are characteristics of public companies. Higher rates of mental health and absenteeism are determined among the employees in public companies. Higher levels of mental issues among private sector employees may be affected by demanding schedules, high stress levels, and achievement orientation in the private sector [63]. The results of multivariate tests indicate the significance of differences between the employees that work in different companies in different sectors and the expression of mental health, job characteristics, and absenteeism [23,29,30,59]. Specifically, the results indicate that performing a job in the health sector includes a wider range of jobs and skills that are required for its performance, compared to jobs in the administration sector. When it comes to the production sector, feedback on the success of work performance is a job characteristic that is significantly more pronounced compared to the administration and health sectors. The successful performance of the work of employees in the health and production sectors is more determined by joint cooperation and synergy of knowledge, skills, and capacities of employees than in the administration sector. In the production sector, the friendship component is more expressed than in administration. Finally, when it comes to mental health, according to the results obtained, employees in the hospitality industry showed fewer mental health issues compared to employees from the healthcare and production sectors. Moreover, there are certain indicators that employees in the hospitality industry, as well as in administration, are better adapted to existing working conditions.

## 5. Conclusions

This study applied the quantitative method to investigate the factors affecting absenteeism. The specific factors that were considered in explaining absenteeism were mental health and job characteristics among the employees from organizations of different sizes, ownership, and sectors. The findings suggest that the mental health of employees represents a significant and unequivocal prerequisite in reducing absenteeism rates in the limited working conditions created under the influence of Industry 4.0 and the circumstances of the COVID-19 pandemic. Moreover, certain job characteristics in organizations of different types can significantly reduce the rates of absenteeism. The results support the premises of Industry 5.0 and offer a new human-centric approach to absenteeism through the promotion of mental health through long-term organizational strategies and a more inclusive approach to employee preferences in relation to job characteristics. Regardless of the size, type of ownership, or sector, the HRM organization must permanently strengthen human capacities and improve their working conditions. The research determined the significant group of predictors in terms of absenteeism. According to the results, mental health and job characteristics can be used as a starting premise for job design in order to reduce the absence of employees in accordance with the results given for different types of companies. Absenteeism is not an isolated phenomenon. It is always affected by personal and organizational factors, so it is important for organizations to manage all processes and resources in a way that would reduce the potential for absenteeism. The main limitation of the study is the sample size and the territory covered by the research. Future studies should replicate the model among countries in development in order to determine whether there are similar tendencies in the factors that affect absenteeism among employees who work in the organizations of different types. Moreover, the current study is investigating only the effects of different characteristics of organizations; it is necessary to investigate the effects of the sociodemographic characteristic of the employees when it comes to absenteeism. In the end, future studies should provide evidence for the effects of implementing the HRM programs that promote mental health and the employee’s fit within the working environment, which will contribute to a more human-centric, inclusive, and sustainable working environment.

## Figures and Tables

**Figure 1 behavsci-13-00332-f001:**
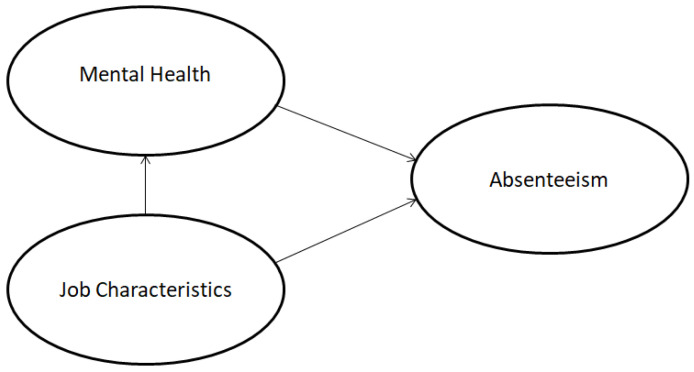
Conceptual framework.

**Table 1 behavsci-13-00332-t001:** The annual costs associated with absenteeism among different occupations.

Occupation	Annual Cost of Lost Productivity Due to Absenteeism (in Billions)
Professional	USD 24.2
Managers/executives	USD 15.7
Service workers	USD 8.5
Clerical/office	USD 8.1
Sales	USD 6.8
School teachers	USD 5.6
Nurses	USD 3.6
Transportation	USD 3.5
Manufacturing/production	USD 2.8
Business owners	USD 2.0
Installation/repair	USD 1.5
Construction/mining	USD 1.3
Physicians	USD 0.25
Farmers/foresters/fishers	USD 0.16

**Table 2 behavsci-13-00332-t002:** Characteristics of the sample (N = 502).

Variable	Category	n	%
Gender	Male	184	36.7
	Female	318	63.3
Age	Range	18–65	
	M(SD)	40.14 (11.17)	
Education	High School	197	39.2
	Higher School	33	6.6
	Bachelor	171	34.1
	Master	88	17.5
	PhD	13	2.6
Ownership	Private	152	30.3
	Public	350	69.7
Sector	Hospitality	97	19.3
	Health	103	20.5
	Manufacturing	88	17.5
	Administration	214	42.6
Size	Small	105	20.9
	Medium	107	21.3
	Large	290	57.8

**Table 3 behavsci-13-00332-t003:** Descriptive statistics and the reliability of scale dimensions.

Variables	Min	Max	M	SD	Sk	Ku	α
Mental health	1.20	6.00	3.89	0.833	−0.443	−0.558	0.796
Job variety	1.60	5.00	3.98	0.613	−0.510	0.278	0.677
Autonomy	1.00	5.00	3.48	0.716	−0.164	−0.200	0.627
Feedback	1.00	5.00	3.73	0.873	−0.504	−0.352	0.820
Dealing with others	1.67	5.00	3.96	0.720	−0.708	0.285	0.689
Task Identity	1.25	5.00	3.98	0.753	−0.676	0.194	0.663
Friendship	1.57	5.00	4.08	0.704	−0.650	−0.004	0.810
Absenteeism	0.00	54.00	24.34	0.523	−0.353	−0.654	−

**Table 4 behavsci-13-00332-t004:** The prediction of absenteeism in the context of job characteristics and mental health among different sectors.

	Hospitality			Health			Production			Administration		
	*β*	*t*	*p*	*β*	*t*	*p*	*β*	*t*	*p*	*β*	*t*	*p*
Job variety	0.21	1.638	0.105	−0.051	−0.472	0.638	−0.10	−0.746	0.458	0.13	1.789	0.075
Autonomy	0.28	1.959	0.046	−0.037	−0.274	0.785	0.08	0.541	0.590	0.05	0.649	0.517
Feedback	0.14	1.178	0.242	0.006	0.046	0.963	0.13	1.146	0.255	−0.15	−1.662	0.044
Dealing with others	−0.10	0.729	0.468	0.100	0.741	0.461	0.01	0.013	0.990	−0.05	−0.613	0.540
Task Identity	−0.11	0.867	0.388	0.148	1.176	0.242	−0.13	−0.864	0.390	0.12	1.458	0.146
Friendship	−0.03	0.185	0.854	−0.103	−0.720	0.474	−0.34	−2.670	0.009	0.08	0.880	0.380
MHI	−0.31	−2.112	0.039	−0.22	−1.984	0.049	−0.32	−2.577	0.012	0.19	2.431	0.032

**Table 5 behavsci-13-00332-t005:** Post hoc tests—significant differences between the organizations of different size.

Variable	(I) Size (M)	(J) Size (M)	(I-J)	SE	*p*
Job variety	Large (4.07)	Small (3.86)	0.21 *	0.069	0.012
		Medium (3.86)	0.21 *	0.068	0.012
Dealing with others	Large (4.05)	Small (3.84)	0.21 *	0.081	0.030
		Medium (3.83)	0.22 *	0.081	0.023
Friendship	Large (4.15)	Medium (3.95)	0.20 *	0.079	0.040
Mental health	Large (3.80)	Small (4.11)	0.31 *	0.094	0.005

* mean difference.

**Table 6 behavsci-13-00332-t006:** *t*-test on independent samples—difference in ownership.

	Ownership	N	M	SD	*t*	*p*
Job variety	Private	152	4.01	0.664	0.765	0.444
	Public	350	3.97	0.589		
Autonomy	Private	152	3.58	0.741	2.091	0.037
	Public	350	3.44	0.701		
Feedback	Private	152	3.82	0.869	1.535	0.126
	Public	350	3.69	0.873		
Dealing with others	Private	152	4.08	0.750	2.467	0.014
	Public	350	3.91	0.701		
Task Identity	Private	152	3.95	0.791	0.613	0.540
	Public	350	3.93	0.736		
Friendship	Private	152	4.21	0.691	2.831	0.005
	Public	350	4.02	0.699		
Mental health	Private	152	3.72	0.870	2.916	0.004
	Public	350	3.96	0.807		
Absenteeism	Private	152	25.22	1.211	−3.141	0.001
	Public	350	32.35	1.114		

**Table 7 behavsci-13-00332-t007:** Post hoc tests—significant differences between the organizations of different sector.

Variable	(I) Sector	(J) Sector	(I-J)	S.E	*p*
Job variety	Health (4.11)	Administration (3.90)	0.21 *	0.072	0.043
Feedback	Production (3.94)	Health (3.39)	0.55 *	0.124	0.000
	Administration (3.82)	Health (3.39)	0.43 *	0.102	0.001
Dealing with others	Health (4.10)	Administration (3.84)	0.26 *	0.084	0.024
	Production (4.21)	Hospitality (3.85)	0.36 *	0.103	0.009
		Administration (3.84)	0.37 *	0.089	0.001
Friendship	Production (4.31)	Administration (3.99)	0.32 *	0.088	0.004
Mental health	Hospitality (4.04)	Health (3.67)	0.37 *	0.114	0.020
		Production (3.56)	0.48 *	0.118	0.001
	Administration (4.06)	Health (3.67)	0.39 *	0.096	0.002
		Production (3.56)	0.50 *	0.102	0.000

* mean difference.

## Data Availability

Not applicable.

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
