# Peer review of "The Psychosocial Model of Absenteeism: Transition from 4.0 to 5.0"

_behavsci, 2023, doi:10.3390/bs13040332_

Round 1
Reviewer 1 Report
The paper is written well and the statistics are clear. I have three questions, which I am sharing with the editors. 1) the link between mental health and absenteeism is well-established, and this paper didn't add to that existing knowledge. Perhaps if there was more discussion about how this fits into 5.0 that might merit value added, or if this measure of mental health differs from previously used measures, or if the differences between professions was central. 2) an average of 24 absentee days in the last 12 months (page 6, line 234) is quite significant, it raises serious questions about the representativeness or appropriateness of the sample. 3) the paper lacks any presentation of how the sample was recruited or the survey administered. My guess is this was an internet survey but how were the participants selected and provided the link to the survey?
Author Response
Dear Reviewer,
I am writing to submit the revised version of our manuscript entitled The psychosocial model of absenteeism: Transition from 4.0 to 5.0, which you kindly reviewed for us.
Firstly, I would like to express my sincere appreciation for your valuable comments and feedback on our manuscript. Your insights and suggestions have been helpful in improving the quality of our work, and we are grateful for your time and effort in reviewing our manuscript.
We have carefully addressed all the comments and suggestions raised by you and have made the necessary revisions to the manuscript (Please see the attachment). We hope that you will find our revised manuscript satisfactory and suitable for publication.
Once again, thank you for your valuable input and for considering our manuscript for publication. We look forward to hearing your feedback on the revised manuscript.
1) the link between mental health and absenteeism is well-established, and this paper didn't add to that existing knowledge. Perhaps if there was more discussion about how this fits into 5.0 that might merit value added, or if this measure of mental health differs from previously used measures, or if the differences between professions was central – Please find attach – track changes
2) an average of 24 absentee days in the last 12 months (page 6, line 234) is quite significant, it raises serious questions about the representativeness or appropriateness of the sample. -According to the Labor Law in the Republic of Serbia, there are as many as 13 reasons for absence (Please find attach- track changes). Only on holidays and religious celebrations, employees are absent on average about 30 days per year.
3) the paper lacks any presentation of how the sample was recruited or the survey administered. My guess is this was an internet survey but how were the participants selected and provided the link to the survey? Please find attach - track changes
Sincerely, Aleksandra Sitarević

Reviewer 2 Report
I have received an article titled “The psychosocial model of absenteeism: Transition from 4.0 to 2 5.0”. It is an interesting topic. However, following suggestions may help improve the article;
1. The significance of the study may please be added to the abstract.
2. The novelty of the study may please be added to the abstract section.
3. It is advisable to add some statistical figures in the introduction section showing the loss of companies (in Dollars) due to the unplanned absenteeism.
4. Literature is adequately reported.
5. The research framework diagram is missing.
6. How the authors avoided the self-reported bias?
7. Discussion must cater for the results of this study in comparison to the earlier studies.
8. Thank you
Author Response
Dear Reviewer,
I am writing to submit the revised version of our manuscript entitled The psychosocial model of absenteeism: Transition from 4.0 to 5.0, which you kindly reviewed for us.
Firstly, I would like to express my sincere appreciation for your valuable comments and feedback on our manuscript. Your insights and suggestions have been helpful in improving the quality of our work, and we are grateful for your time and effort in reviewing our manuscript.
We have carefully addressed all the comments and suggestions raised by you and have made the necessary revisions to the manuscript (Please see the attachment). We hope that you will find our revised manuscript satisfactory and suitable for publication.
Once again, thank you for your valuable input and for considering our manuscript for publication. We look forward to hearing your feedback on the revised manuscript.
Reviewer 2
I have received an article titled “The psychosocial model of absenteeism: Transition from 4.0 to 2 5.0”. It is an interesting topic. However, following suggestions may help improve the article;
- The significance of the study may please be added to the abstract. - Please find attach - track changes
- The novelty of the study may please be added to the abstract section. - Please find attach - track changes
- It is advisable to add some statistical figures in the introduction section showing the loss of companies (in Dollars) due to the unplanned absenteeism. - Please find attach - track changes
- Literature is adequately reported. - OK
5.The research framework diagram is missing. - Please find attach - track changes
- How the authors avoided the self-reported bias? - Please find attach - track changes
- Discussion must cater for the results of this study in comparison to the earlier studies. - Please find attach - track changes
- Thank you - Thank you
Sincerely, Aleksandra Sitarević
